# Efficient mRNA delivery to resting T cells to reverse HIV latency

Paula M. Cevaal [1,11], Stanislav Kan[1,11], Bridget M. Fisher [1,11], Michael A. Moso [1,2,11], Abigail Tan[1], Haiyin Liu[3], Abdalla Ali[1], Kiho Tanaka [1], Rory A. Shepherd [1], Youry Kim[1], Jesslyn Ong[1], Denzil L. Furtado[4], Marvin Holz[5], Damian F. J. Purcell [5], Joshua M. L. Casan [6,7], Thomas Payne[3], Wei Zhao[1], Mohamed Fareh[6,7], James H. McMahon [8], Steven G. Deeks [9], Rebecca Hoh[9], Sushama Telwatte[1], Colin W. Pouton [3], Angus P. R. Johnston [3], Frank Caruso [4], Jori Symons [10], Sharon R. Lewin [1,2,8,12] ✉ & Michael Roche[1,12]

A major hurdle to curing HIV is the persistence of integrated proviruses in resting CD4[+] T cells that remain in a transcriptionally silent, latent state. One strategy to eradicate latent HIV is to activate viral transcription, followed by elimination of infected cells through virus-mediated cytotoxicity or immune-mediated clearance. We hypothesised that mRNA-lipid nanoparticle (LNP) technology would provide an opportunity to deliver mRNA encoding proteins able to reverse HIV latency in resting CD4[+] T cells. Here we develop an LNP formulation (LNP X) with unprecedented potency to deliver mRNA to hard-to-transfect resting CD4[+] T cells in the absence of cellular toxicity or activation. Encapsulating an mRNA encoding the HIV Tat protein, an activator of HIV transcription, LNP X enhances HIV transcription in ex vivo CD4[+] T cells from people living with HIV. LNP X further enables the delivery of clustered regularly interspaced short palindromic repeats (CRISPR) activation machinery to modulate both viral and host gene transcription. These findings offer potential for the development of a range of nucleic acid-based T cell therapeutics.

Recent advances in mRNA and lipid nanoparticle (LNP) technology have allowed for the development of new vaccines and therapeutics, holding great promise for gene therapy. In 2018, patisiran (Onpattro) was approved as the first LNP-based therapeutic for the delivery of silencing (si)RNA for transthyretin-mediated amyloidosis[1]. The same platform has since been used to deliver mRNA to generate two of the most effective vaccines against COVID-19 (Comirnaty (Pfizer/ BioNTech) and Spikevax (Moderna)). More recently, LNPs have been used to deliver mRNA for CRISPR-Cas9 gene editing in vivo, which was found to be safe in human clinical trials[2]. These advances have triggered enormous interest in the use of mRNA-LNPs as versatile therapeutics, including for infectious diseases such as HIV.

Whilst antiretroviral therapy (ART) effectively inhibits active HIV replication and reduces morbidity and mortality, it is not curative and

[1]Department of Infectious Diseases, The University of Melbourne at The Peter Doherty Institute for Infection and Immunity, Melbourne, VIC, Australia. [2]Victorian Infectious Diseases Service, The Royal Melbourne Hospital at the Peter Doherty Institute for Infection and Immunity, Melbourne, VIC, Australia. [3]Monash Institute of Pharmaceutical Sciences, Monash University, Parkville, VIC, Australia. [4]Department of Chemical Engineering, The University of Melbourne, Parkville, VIC, Australia. [5]Department of Microbiology and Immunology, The University of Melbourne at The Peter Doherty Institute for Infection and Immunity, Melbourne, VIC, Australia. [6]Cancer Immunology Program, Peter MacCallum Cancer Centre, Melbourne, VIC, Australia. [7]Sir Peter MacCallum Department of Oncology, The University of Melbourne, Parkville, VIC, Australia. [8]Department of Infectious Diseases, Alfred Hospital and Monash University, Melbourne, VIC, Australia. [9]Department of Medicine, University of California, San Francisco, San Francisco, CA, USA. [10]Translational Virology, Department of Medical Microbiology, University Medical Center, Utrecht, the Netherlands. [11]These authors contributed equally: Paula M. Cevaal, Stanislav Kan, Bridget M. Fisher, Michael A. Moso. [12]These authors jointly supervised this work: Sharon R. Lewin, Michael Roche. ✉e-mail: sharon.lewin@unimelb.edu.au

treatment is life-long[3]. The major barrier to HIV cure is the persistence of latently infected, resting CD4[+] T cells harbouring replication-competent virus[4-6], which can rebound following T cell activation and re-establish viremia in the absence of ART. One approach towards an HIV cure is to reactivate HIV transcription using latency-reversing agents (LRAs) while on ART, with the goal of subsequently inducing death of the infected cells and reduction of the HIV reservoir through viral cytopathic effects or immune-mediated clearance of the infected cell[7,8]. Traditionally, this has been achieved using small, hydrophobic compounds, that target different cellular pathways to activate transcription[9,10]. These first-generation LRAs, such as histone deacetylase inhibitors, can be administered systemically and passively cross the plasma membrane to exert their effect intracellularly. Several LRAs have demonstrated induction of HIV RNA in vitro, ex vivo and in clinical trials; yet, to date, no clinical trial of an LRA alone has shown a reduction in the size of the HIV reservoir[7,8,11-15].

There are several hypotheses as to why LRAs alone have not been able to induce clearance of infected cells. First, most first-generation LRAs only increase the initiation of HIV transcription (as measured by unspliced HIV RNA), but fail to overcome subsequent blocks in transcription elongation, completion and splicing that persist in a resting CD4[+] T cell[10]. Second, first-generation LRAs are not HIV-specific, and therefore their potency to reactivate HIV is unavoidably coupled to off-target and adverse effects[8,16,17]. Indeed, certain LRAs have been shown to directly inhibit CD8[+] T cell and natural killer cell function in vitro, potentially hampering immune-mediated clearance of reactivated cells[18,19]. Therefore, there is a need for new LRAs that have greater potency, lower toxicity, and greater specificity for the HIV provirus.

Recently, two HIV-specific nucleic acid-based LRAs have been reported. The first is an LNP encapsulating mRNA encoding the HIV protein Trans-activator of Transcription (Tat), which binds to the trans-activation response element (TAR) in nascent HIV transcripts produced from the HIV long terminal repeat (LTR) promotor region and potently enhances transcriptional processivity[20-22]. Although this Tat-LNP could reverse latency both in vitro and ex vivo[21,22], a small molecule non-specific LRA was required to achieve high potency—an observation consistent with suboptimal delivery of the Tat mRNA[21]. The second is HIV LTR-targeted CRISPR activation (CRISPRa), comprising of a catalytically inactive Cas9 protein combined with transcriptional activator domains, which provides a promising strategy for highly HIV-specific activation of transcription without affecting host-cell transcription[23-28]. However, both these novel LRAs have not progressed to primary cells or the clinic due to the lack of an efficient delivery vehicle to resting CD4[+] T cells, which are known to be recalcitrant to traditional gene delivery systems[29].

We therefore aimed to develop an LNP capable of delivering mRNA to resting T cells for potent, HIV-specific latency reversal. We identified an LNP formulation with the unique capability to transfect CD4[+] T cells in the absence of pre-stimulation. We then used this LNP formulation, LNP X, to encapsulate two mRNA-based LRAs encoding for HIV Tat and CRISPRa, which demonstrated efficient activation of HIV transcription in CD4[+] T cells from people living with HIV on ART ex vivo. Together, these findings provide an exciting new approach to mRNA-based therapeutics for T cells.

## Results

### LNP X enables delivery of mRNA to resting T cells

We first assessed whether an LNP similar to an existing, FDA-approved LNP formulation (patisiran) was able to transfect non-stimulated primary CD4[+] T cells by delivering a reporter mCherry mRNA. Our patisiran-like LNP (hereafter referred to as patisiran LNP) is formulated with DMG-PEG2000 instead of PEG2000-C-DMG. At the highest dose tested of 500 ng per 10[5] cells, mCherry expression was only detected in 2.1 ± 0.4% (mean ± SEM) of live cells (Fig. 1a and Fig. S1a) after 72 h incubation. Pre-stimulation of the CD4[+] T cells with anti-CD3/anti-

CD28 led to substantially higher transfection efficiencies of up to 51 ± 5.1% compared to non-stimulated T cells (Fig. 1a), though the toxicity associated with patisiran LNP treatment was greater in pre-stimulated compared to non-stimulated CD4[+] T cells at higher LNP doses (Fig. S1b). These data show that successful mRNA delivery to primary CD4[+] T cells using patisiran LNPs is dependent on the activation state of the T cell.

We therefore modified the lipid composition of the LNP to enhance potency. First, the ionisable lipid DLin-MC3-DMA (MC3) was replaced with SM-102, an ionisable lipid previously shown to lead to greater cytosolic mRNA delivery through enhanced endosomal escape[30]. Second, the SM-102-LNPs were further modified using ß-sitosterol, a naturally-occurring cholesterol analogue associated with enhanced mRNA delivery[31], to create a formulation referred to as LNP X (Fig. 1b). LNP X encapsulating reporter mCherry mRNA was formulated reproducibly and did not significantly differ in size, polydispersity or mRNA encapsulation efficiency compared to patisiran LNP (Fig. 1c and Table 1).

We assessed the potency of LNP X by treating Jurkat T cells, an immortalised T cell line, with increasing doses of reporter mRNA encapsulated in either patisiran LNP or LNP X. Treatment with LNP X compared to patisiran LNP resulted in up to 6-fold higher mCherry expression (Fig. 1d), confirming its superior potency. To assess the potency of our LNP X formulation further, we treated pre-stimulated CD4[+] T cells with LNP X and observed mCherry expression in >75% of cells, in all doses tested, including a low dose of 6.25 ng mCherry-LNP X per 10[5] cells (Fig. 1e and Fig. S1a). Impressively, and in stark contrast to patisiran LNP, LNP X was able to achieve transfection of up to 76 ± 3.8% CD4[+] T cells in the absence of pre-stimulation (Fig. 1a–e). Moreover, toxicity was negligible compared to baseline in both pre-stimulated and non-stimulated CD4[+] T cells (Fig. S1c). To our knowledge, this is the first demonstration of potent transfection of primary, resting T cells in vitro in the absence of cellular toxicity.

LNP X was able to transfect both naïve and all memory T cell subsets (Fig. S2a–d), with effector memory T cells expressing higher levels of mCherry than naïve T cells ($p = 0.007$), potentially reflecting the higher basal translational activity in this memory T cell subset. Even in the context of peripheral mononuclear cells (PBMCs), LNP X was able to deliver mRNA to CD4[+] T cells (Fig. S2e, f). Expression of mRNA was also detected in most other PBMC subsets, specifically monocytes, demonstrating that LNP X is T cell tropic, but not T cell specific.

### Superior potency of LNP X is not related to endosomal escape

To understand the mechanism behind the superior potency of LNP X, we employed a SNAP_switch reporter system to quantify sub-cellular localisation of therapeutics in Jurkat T cells[32] (Fig. 2a). SNAP_switch was recently used to probe the role that ionisable lipids play in delivering mRNA to the cytosol. This work showed that SM-102 had similar cytosolic delivery compared to DLin-MC3-DMA, the ionisable lipid found in patisiran LNP, however the protein expression induced by SM-102 was significantly higher[33]. To probe LNP X's superior potency, we formulated SNAP_switch LNP X and compared these to LNPs formulated with SM-102, DSPC, DMG-PEG2000 and cholesterol. SNAP_switch LNP X did not exhibit different physicochemical characteristics compared to LNP X encapsulating a single reporter mRNA (Fig. S3). The LNP X formulation resulted in 10.5-fold enhanced mRNA expression in SNAP_switch Jurkat T cells ($p = 0.0029$, Fig. 2b). This enhanced transfection was in part explained by a 2.5-fold increase in LNP association, representing LNPs bound to the cell surface or internalised ($p = 0.0025$, Fig. 2c), yet LNP X still exhibited 4.1-fold higher protein expression relative to the amount of LNP association ($p = 0.0032$, Fig. 2d).

While LNP X yielded an overall higher level of cytosolic cargo delivery (Fig. 2e), this effect correlated with the higher levels of LNP association to cells observed with LNP X. Indeed, the efficiency of

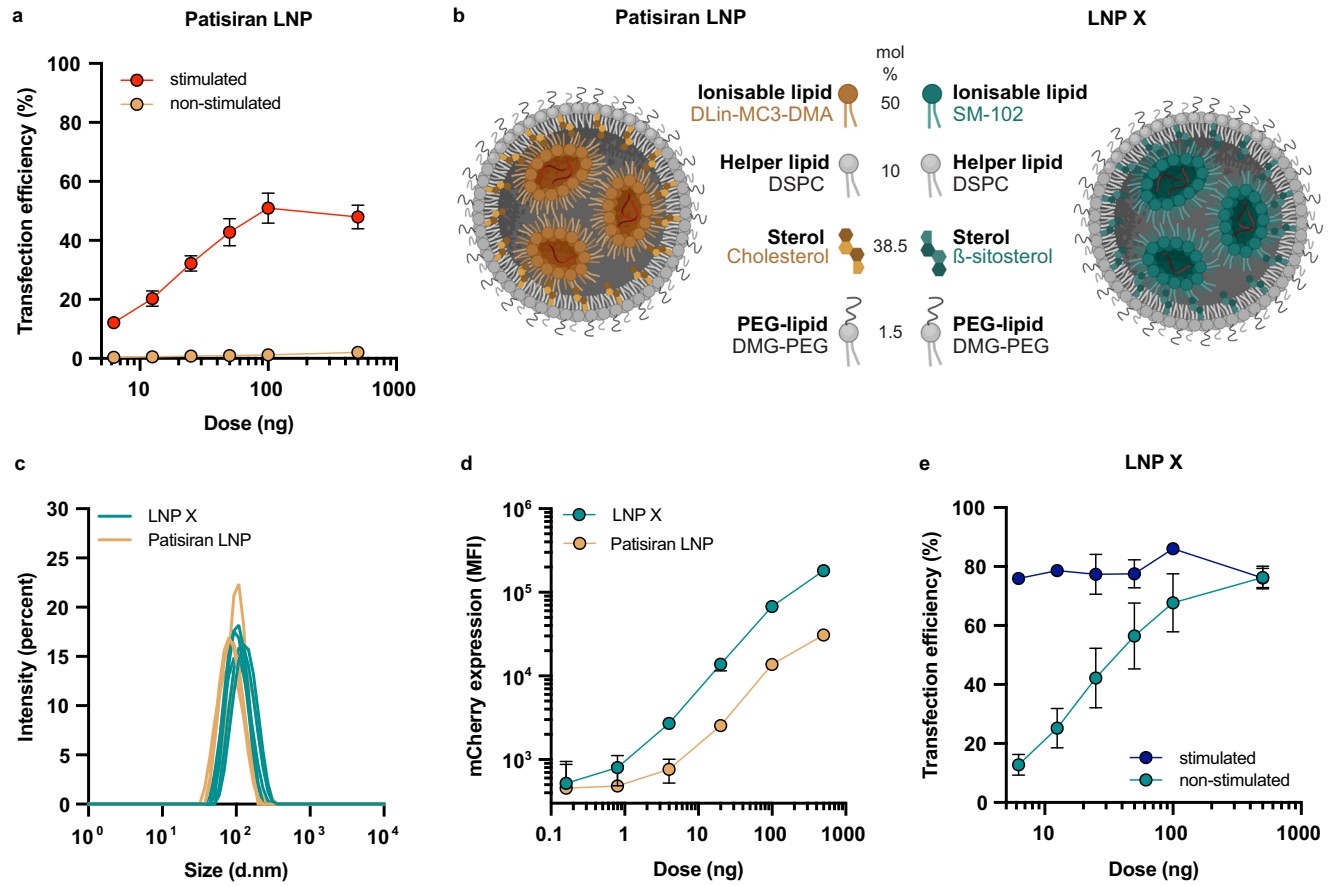

**Fig. 1 | LNP X formulation potently transfects primary CD4⁺ T cells in the absence of pre-stimulation. a** CD4⁺ T cells from HIV-negative donors were rested or pre-stimulated with anti-CD3/anti-CD28, then treated with LNPs (similar to the patisiran lipid composition) encapsulating mCherry mRNA for 72 h at the indicated doses. Highest dose of mCherry-LNP X corresponds to 2.5 µg/mL. Transfection efficiency was determined by measuring mCherry expression using flow cytometry. Mean ± SEM, *n* = 6 donors. **b** Schematic representation of patisiran-like LNP formulation *versus* novel LNP formulation X. Mol %, relative molar percentage of indicated lipid in lipid mixture. Created in BioRender: *Cevaal, P.* (2025) https://

BioRender.com/ypaicaz. **c** Size distribution of patisiran LNP (yellow) *versus* LNP X (green) encapsulating mCherry mRNA. Lines represent individual LNP batches. **d** Relative mCherry expression levels in Jurkat T cells treated for 24 h with indicated doses of patisiran LNP (yellow) or LNP X (green) encapsulating mCherry mRNA. MFI, median fluorescent intensity. Mean ± SEM, n = 2. **e** CD4⁺ T cells from HIV-negative donors were rested or pre-stimulated with anti-CD3/anti-CD28, then treated with LNP X encapsulating mCherry mRNA for 72 h at the indicated doses. Highest dose of mCherry-LNP X corresponds to 2.5 µg/mL. Transfection efficiency was determined using flow cytometry. Mean ± SEM, *n* = 6 donors.

endosomal escape was not improved by replacing cholesterol with ß-sitosterol (Fig. 2f). Importantly, the protein expression relative to the amount of mRNA delivered to the cytosol was enhanced by 5.3-fold when using LNP X (*p* = 0.0012, Fig. 2g), suggesting that the superior potency of LNP X involved processes downstream of endosomal escape and cytosolic delivery of the mRNA cargo. Our findings regarding the role of ß-sitosterol are consistent with previous studies comparing DLin-MC3-DMA/cholesterol to DLin-MC3-DMA/ß-sitosterol

LNPs[33], indicating the observed effects of ß-sitosterol are independent of the ionisable lipid.

### LNP X delivers mRNA encoding HIV Tat to reverse HIV latency
We next aimed to assess whether LNP X was able to deliver a small, therapeutic mRNA expressing the 72 amino acids of the first coding exon of HIV Tat (Tat-LNP X, Fig. 3a and Table S1). This form of HIV Tat is normally expressed from the late-phase, Rev-dependent 4 kb viral Tat mRNA[34]. We first validated the potency of Tat-LNP X in J-Lat 10.6 cells, a T cell line model of HIV latency that expresses green fluorescent protein (GFP) under the control of the HIV LTR promoter. Treatment with Tat-LNP X, but not control mCherry-LNP X, resulted in an increase in GFP expression consistent with potent reactivation of LTR-mediated transcription (Fig. 3b). No impact on cellular viability was observed (Fig. S4). Similarly, Tat-LNP X was able to induce productive infection in primary CD4⁺ T cells infected with an HIV reporter virus (pMorpheus-V5[35]; *p* = 0.0025) similar to levels seen after T cell activation with the mitogens phorbol 12-myristate 13-acetate (PMA) and ionomycin (Fig. S5).

To assess latency reversal ex vivo, we treated CD4⁺ T cells from people living with HIV on suppressive ART with an equivalent dose of 200 ng per 10⁵ cells Tat-LNP X or control mCherry-LNP X for 48–72 h. The expression of cell-associated HIV RNA was measured as described

### Table. 1 | Characterisation of patisiran LNP and LNP X

|  | Patisiran LNP | LNP X | p value |
|---|---|---|---|
| Z-average (d.nm) | 90.8 ± 6 | 105.8 ± 5 | 0.110 |
| Size by Number distribution (d.nm) | 61.2 ± 5 | 73.6 ± 4 | 0.112 |
| PDI | 0.10 ± 0.02 | 0.10 ± 0.01 | 0.995 |
| Encapsulation efficiency (%) | 96.2 ± 6 | 96.4 ± 2 | 0.847 |

Z-average (hydrodynamic diameter), mean size determined by Number distribution and LNP size uniformity (polydispersity index, PDI) as determined by dynamic light scattering for the patisiran LNP or LNP X formulation encapsulating mCherry mRNA. mRNA encapsulation efficiency for both formulations was determined using a modified RiboGreen assay. Mean ± SEM for *n* = 4-5 LNP batches for each formulation. *P* values determined using two-tailed unpaired t test.

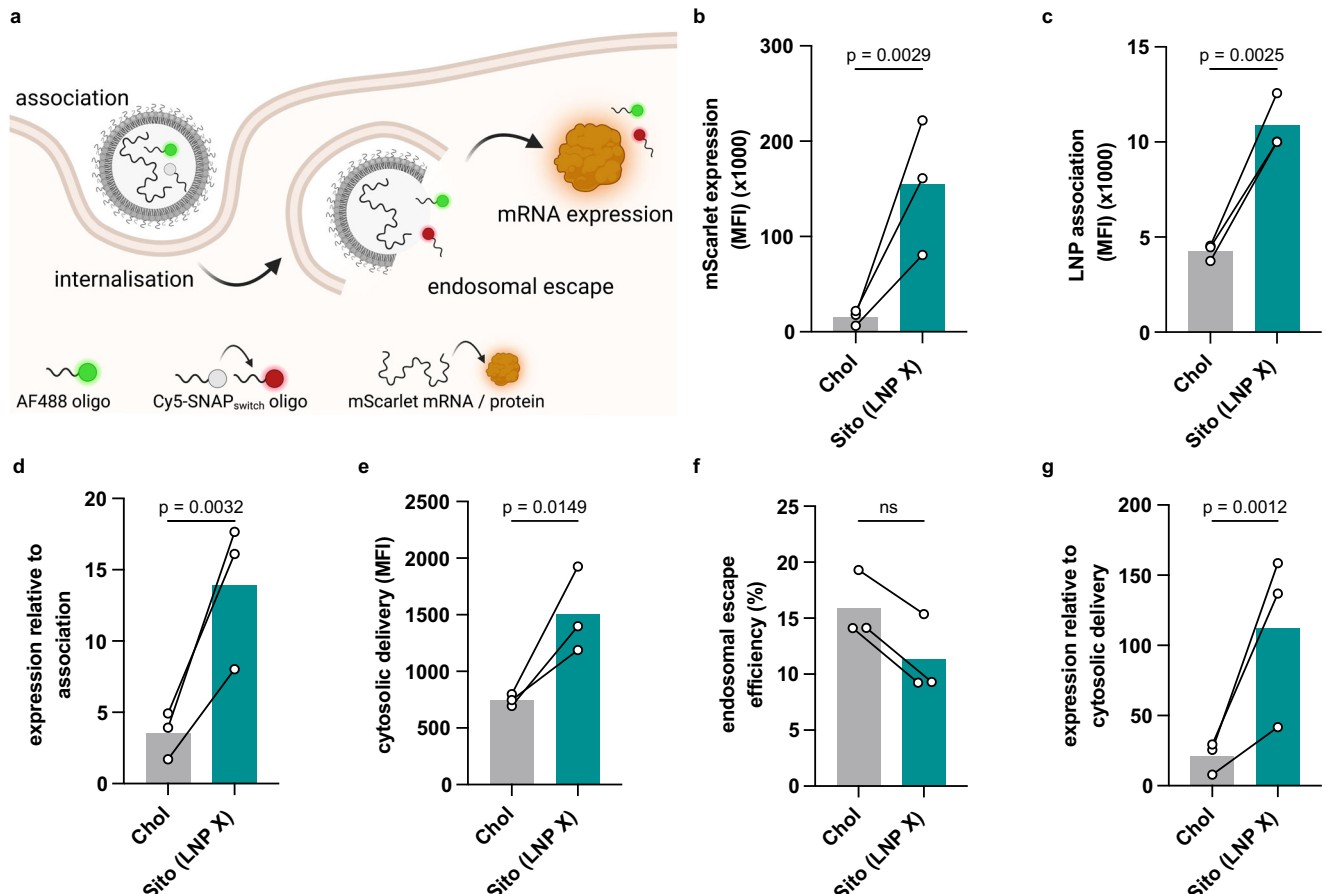

**Fig. 2 | Superior potency of LNP X is not explained by differences in endosomal escape. a** The SNAP$_{switch}$ assay simultaneously tracks the degree of nanoparticle association, endosomal escape and mRNA expression of LNPs co-encapsulating an AF488 oligo, Cy5-SNAP$_{switch}$ oligo and mRNA encoding mScarlet. Figure adapted from Liu et al.[33] and created in BioRender: *Cevaal, P. (2025)* https://BioRender.com/3ajxnam. **b–g** SNAP$_{switch}$-reporter Jurkat T cells were incubated with LNPs containing SM-102, DSPC, DMG-PEG2000 and either Cholesterol (Chol) or ß-sitosterol (Sito; LNP X) for 4 h. LNPs encapsulated a reporter mScarlet mRNA, AF488-tagged oligo and a Cy5-SNAP$_{switch}$ oligo or constitutively active Cy5-oligo. Fluorescence was determined using flow cytometry. **b** Protein expression of the mScarlet reporter mRNA. **c** LNP association as determined by the fluorescence of the AF488-tagged oligo. **d** mScarlet protein expression (as in **b**) relative to LNP association (as in **c**). **e** Amount of cytosolic delivery of nucleic acid cargo as quantified by the Cy5-SNAP$_{switch}$ oligo fluorescence. **f** Efficiency of endosomal escape based on the Cy5-SNAP$_{switch}$ oligo fluorescence (as in **e**) normalised to a constitutively fluorescent Cy5-tagged oligo. **g** mScarlet protein expression (as in **b**) relative to the amount of cytosolic cargo delivery (as in **e**). **b–g** Bars represent aggregate mean of $n = 3$ independent experiments, each symbol representing the average of triplicate technical replicates. Significance was determined using a one-tailed paired ratio t test in (**b–e,g**) to allow comparisons of MFI values between experiments, or one-tailed unpaired t test in (**f**), ns non-significant.

previously[36,37], whereby we simultaneously quantified transcription initiation, elongation, completion and splicing[37]. After a single dose of Tat-LNP X, all measured HIV transcripts were significantly upregulated compared to non-treated control cells suggesting Tat-LNP X induces, and overcomes blocks in, transcription initiation (TAR transcript), proximal and distal elongation (Long-LTR and Pol transcripts, respectively), completion of transcription (Poly(A) transcripts) and splicing (Tat-Rev transcripts; detected by a primer/probe set that spans the junction between the first and second coding exon of *tat* and *rev*) (Fig. 3c–g and Fig. S6a–e). The induction of multiply-spliced Tat-Rev transcripts, an important predictor of virion production ex vivo[38], was induced by 112-fold ($p < 0.01$ compared to untreated control). Treatment with Tat-LNP X or control mCherry-LNP X resulted in minimal overall toxicity (Fig. 3h). Importantly, the induction of HIV RNA transcription following Tat-LNP X significantly exceeded that observed in fully activated T cells stimulated using two mitogens, PMA and phytohaemagglutinin (PHA) (considered the gold-standard to reverse HIV latency in vitro and ex vivo[37,38]) ($p < 0.01$ for all transcripts). However, in contrast to PMA/PHA treatment, Tat-LNP X did not induce any change in cellular activation as observed by minimal changes to

the expression of activation markers CD25, CD69 and HLA-DR compared to non-treated control cells (Fig. S6f–h). Furthermore, treatment with Tat-LNP X, but not mCherry-LNP X, resulted in a 17.2-fold increase in supernatant HIV RNA, indicating virion production following latency reversal (Fig. 3i). However, the reversal of latency by Tat-LNP X did not result in a decline in intact proviral DNA after 5-day culture (Fig. 3j–l). Combined, these findings demonstrate that Tat-LNP X is a highly potent latency-reversing agent that can overcome HIV RNA transcription and processing blocks to elongation, splicing and completion, and induce viral protein expression, all in the absence of T cell activation.

## CRISPR activation machinery can be co-encapsulated by LNP X

We next assessed whether LNP X was able to encapsulate and deliver CRISPR activation (CRISPRa) machinery as a more complex and larger RNA-based, highly HIV-specific therapeutic. The dCas9-synergistic activation mediator (dCas9-SAM) CRISPRa system consists of a catalytically inactive (dead) Cas9 (dCas9) fused to a multimer of C-terminal herpes virus transcriptional activation domain 16 (VP64), which is guided to the genomic target site by a guide RNA (gRNA). The gRNA is

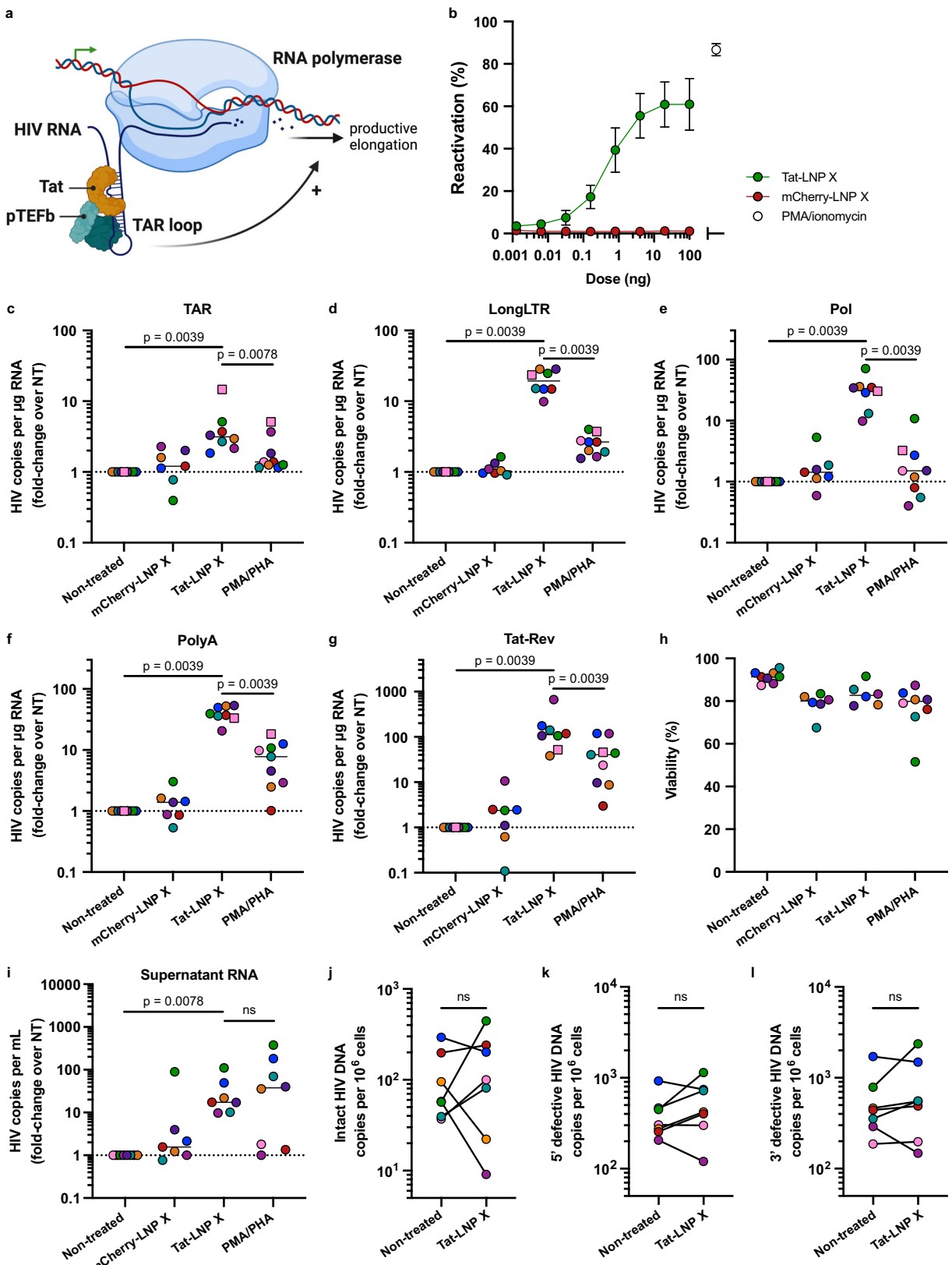

modified to contain minimal hairpin aptamers that allow the recruitment of introduced p65 and heat shock factor 1 (HSF1) transcription activators through binding of the bacteriophage protein MS2[39] (Fig. 4a). The recruitment of multiple copies of the MS2-p65-HSF1 fusion protein, which synergise with dCas9-VP64, greatly enhances the potency of CRISPRa[39], but can result in non-specific promoter

activation if present in excess. We therefore first optimised the relative dose of the three RNAs that comprise the dCas9-SAM CRISPRa machinery. We found that a relative mass ratio of 0.8: 0.00625: 1 (dCas9-VP64 mRNA: MS2-p65-HSF1 mRNA: gRNA) yielded the optimal balance between on-target potency and non-specific background reactivation in cell lines (Fig. S7).

**Fig. 3 | mRNA encoding HIV Tat exon 1 delivered by LNP X is a potent activator of HIV transcription. a** Schematic overview of the mechanism of action of HIV Tat on promoting transcription elongation. TAR; trans-activation response element. Created in BioRender: *Cevaal, P. (2025)* https://BioRender.com/cz40qwv. **b** J-Lat 10.6 cells were treated for 24 h with indicated doses of LNP X encapsulating codon-optimised mRNA expressing the 72 amino acids of the first coding exon of HIV Tat (Tat-LNP X) or mCherry (mCherry-LNP X) as control. Reactivation of HIV LTR-mediated transcription was determined after 24 h by measuring GFP expression. Highest dose of Tat-LNP X corresponds to 1 μg/mL. Treatment with PMA/ionomycin was included as a positive control. Mean ± SEM, $n = 3$ independent experiments. **c–i** CD4$^+$ T cells from people living with HIV on suppressive ART were treated with 200 ng Tat-LNP X or mCherry-LNP X per $10^5$ cells (4 μg/mL) or PMA/PHA as a positive control. After 48 h (squares) or 72 h (circles), expression of HIV transcripts TAR (**c**) LongLTR (**d**), Pol (**e**), PolyA (**f**) and Tat-Rev (**g**) representing transcription initiation, proximal elongation, distal elongation, completion and splicing,

respectively, was determined using digital RT-PCR. Data were normalised to RNA input, then presented as fold-change induction compared to the corresponding non-treated (NT) control. Horizontal dashed line (**c–g**) represents no change relative to untreated cells. **h** Cellular toxicity was determined using flow cytometry. Where datapoints are missing in (**h**), cell input was insufficient to perform an accurate measurement. **i** After 72 h, the number of copies of HIV RNA per mL of supernatant was quantified using RT-PCR and normalised to the non-treated control. **c–i** Short horizontal line represents the median of $n = 7–8$ donors. Significance in (**c–g** and **i**) was determined using a one-tailed Wilcoxon signed-rank test, ns non-significant. After 120 h, the number of intact (**j**), 5′ defective (**k**) or 3′ defective (**l**) HIV DNA copies were determined using digital PCR. Copy numbers were normalised to cell input. Datapoints represent individual donors ($n = 7$). Note different donors were used in (**j–l** and **c–i**). Significance was determined using a two-tailed Wilcoxon signed-rank test, ns non-significant.

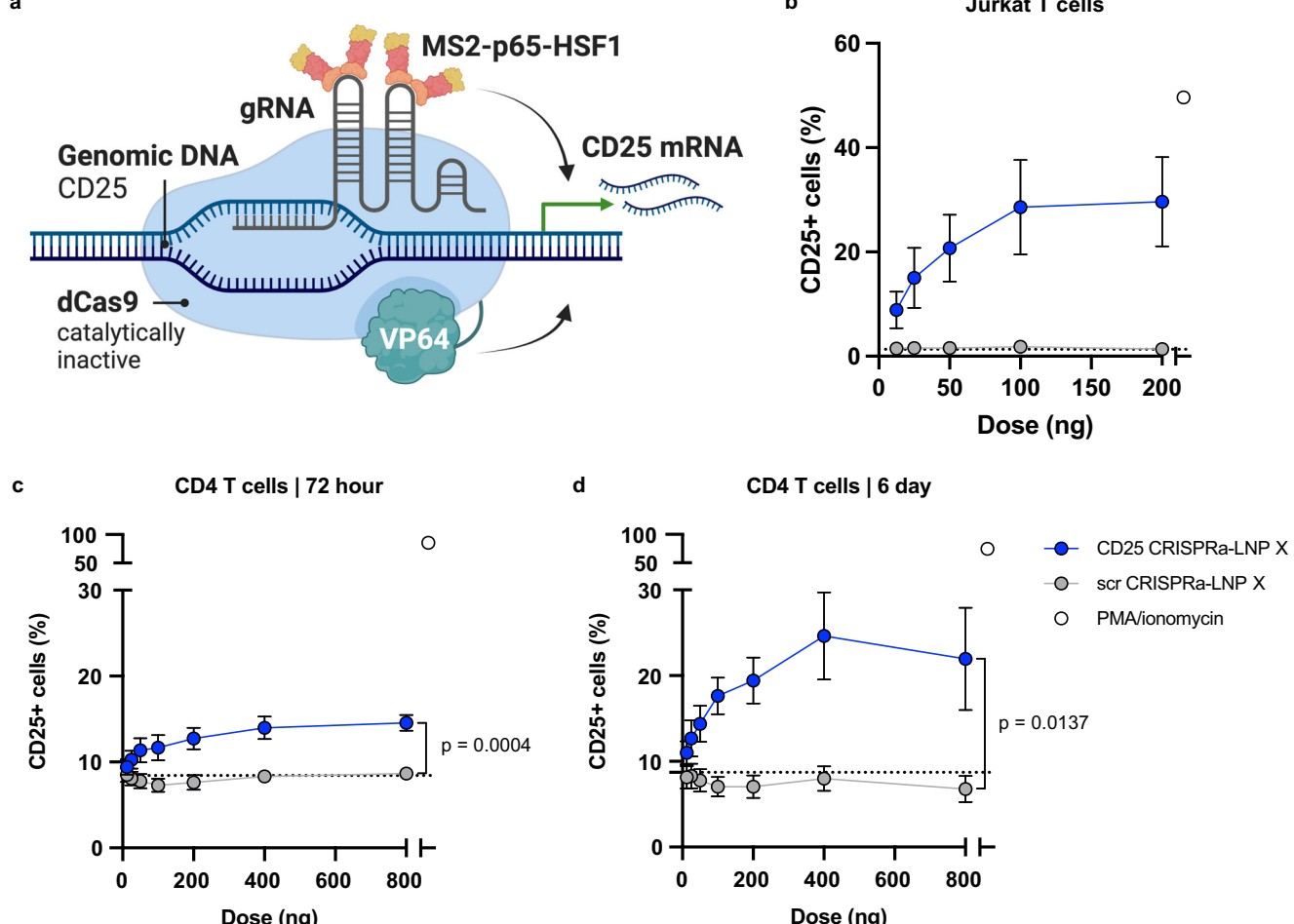

**Fig. 4 | LNP X co-encapsulating CRISPR activation machinery can be used to induce expression of endogenous genes in T cells. a** Schematic overview of the dCas9-synergistic activation mediator (SAM) CRISPR activation system, consisting of a catalytically inactive (dead, d)Cas9 fused to transcriptional activator domain VP64, a gRNA and a further transcriptional activation MS2-p65-HSF1 fusion protein that is recruited to the Cas9-gRNA complex *via* MS2-binding to stem-loop structures in the gRNA scaffold. Created in BioRender: *Cevaal, P. (2025)* https:// BioRender.com/p8j24y6. **b–d** LNP formulation X was used to encapsulate the CRISPR activation machinery (CRISPRa-LNP X) including a gRNA targeting the endogenous gene encoding CD25 (CD25 CRISPRa-LNP X) or a scrambled control

gRNA (scr CRISPRa-LNP X). **b** Jurkat T cells were treated for 24 h with CRISPRa-LNP X, control mCherry-LNP X or PMA/ionomycin as positive control. Induction of CD25 expression was measured by surface stain using flow cytometry. CD4$^+$ T cells from HIV-negative donors were treated with indicated doses of CRISPRa-LNP X per $10^5$ cells or mCherry-LNP X as a control for 72 h (**c**) or 6 days (**d**), after which CD25 expression was determined using flow cytometry. Highest dose of CD25 CRISPRa-LNP X corresponds to 4 μg/mL. Treatment with PMA/ionomycin was included as a positive control. Dotted line represented average baseline CD25 expression in the absence of treatment. Mean ± SEM, $n = 4$ independent experiments (**b**) or $n = 4–6$ donors (**c**, **d**). Significance was determined using a two-tailed Student's t-test.

 

We next developed a model system in which CRISPRa was targeted to the promoter of an endogenous gene, *IL2RA (cd25)*, to induce overexpression of the CD25 receptor on the cell surface. CRISPRa-LNP X was formulated by co-encapsulating the two mRNAs and a CD25-targeting or a scrambled control gRNA. The CD25-targeting gRNA was validated in Jurkat T cells, which express negligible CD25 at baseline. We found that 24 h following treatment with CD25 CRISPRa-LNP X, there was an increase in surface CD25 expression in up to $29.6 \pm 8.6\%$ of Jurkat T cells, while minimal to no background expression of CD25 was observed with scrambled gRNA (scr) CRISPRa-LNP X (Fig. 4b and Fig. S8a). We demonstrated a similar level of potency in non-stimulated primary CD4$^+$ T cells from HIV-negative donors, yielding a significant increase in cells expressing CD25 from $8.7 \pm 0.7\%$ with scr CRISPRa-LNP X to $14.6 \pm 0.9\%$ after 72 h treatment with 800 ng CD25 CRISPRa-LNP X per $10^5$ cells ($p < 0.01$) (Fig. 4c and Fig. S8b).

We then extended the incubation time after a single dose of CD25 CRISPRa-LNP X, scr CRISPRa-LNP X or control mCherry-LNP X from 72 h to 6 days. The percentage of cells expressing CD25 after 6 days was increased to $24.6 \pm 5.1\%$ of live cells (Fig. 4d and Fig. S8c), suggesting the peak response of CRISPRa-induced protein expression occurs later than 72 h post-treatment. In contrast, the expression of mCherry was reduced at 6 days compared to 72 h post-treatment (Fig. S8d, e), consistent with transient expression of LNP-mediated delivery of exogenous mRNA. The enhanced CD25 expression at day 6 was therefore likely not the result of increased expression of the CRISPRa machinery (dCas9-VP64 and MS2-p65-HSF1) at day 6, rather the extended incubation is required to capture the translation of the induced *CD25* mRNA and its trafficking to the cell surface. Importantly, these findings confirm, the first to our knowledge, successful delivery of the dCas9-SAM CRISPRa machinery to non-stimulated T cells in vitro.

## HIV CRISPRa-LNP X activates HIV transcription ex vivo

Finally, to generate HIV LTR-targeted CRISPRa-LNP X, we designed four MS2-p65-HSF1 recruiting hairpin-modified gRNAs (B, C, L and O) targeting different HIV LTR regions upstream of the transcription start site based on previous work[23,40] (Fig. 5a, Table S1 and S2). We found that all four HIV LTR-targeting gRNAs, compared to a scrambled gRNA, could induce GFP expression in J-Lat 10.6 cells (Fig. 5b). To assess the potency of HIV LTR-targeting CRISPRa-LNP X ex vivo, we treated CD4$^+$ T cells from ART-suppressed people living with HIV for 72 h with an equivalent dose of 200 ng CRISPRa-LNP X per $10^5$ cells, analogous to what we used to deliver Tat mRNA. We used CRISPRa-LNP X co-encapsulating a 1:1 mass ratio of gRNA L and gRNA O (L + O CRISPRa-LNP X), which was found to exceed the potency of either gRNA alone (Fig. S9) and was hypothesised to increase the sequence coverage of HIV subtype B LTR sequences, the most common subtype in our cohort. L + O CRISPRa-LNP X treatment significantly increased all measured HIV transcripts except multiply-spliced Tat-Rev (Fig. 5c-g and Fig. S10a-e). The potency of latency-reversal was lower compared to Tat-LNP X treatment, with a maximum fold-increase over non-treated of 2.0 (IQR 0.87-3.39) for Pol transcripts, and did not exceed that of PMA/PHA treatment. No HIV transcripts were upregulated after treatment with a scrambled gRNA CRISPRa-LNP X, indicating the activation of transcription was gRNA-mediated. Treatment with HIV LTR-targeted CRISPRa-LNP X resulted in minimal toxicity, similar to treatment with Tat- or mCherry-LNP X, and did not result in generalised cellular activation (Fig. 5h and Fig. S10f-h), demonstrating the specificity of CRISPRa-LNP X as a next-generation LRA. However, treatment with L + O CRISPRa-LNP X did not lead to an induction of supernatant HIV RNA, as a measure of virion release (Figs. 5i and S10i).

## Discussion

Advances in mRNA technology provide an opportunity for a new generation of therapeutics with enhanced potency and reduced toxicity. Here, we explored the use of LNPs to enable therapeutic mRNA delivery to primary T cells, including CD4$^+$ T cells latently infected with HIV. We reported an LNP formulation that is based on a combination of SM-102 and β-sitosterol. This LNP X reached transfection efficiencies of >75% in primary CD4$^+$ T cells. To our knowledge, this is the first demonstration of successful, non-toxic in vitro transfection of primary T cells in the absence of T cell pre-stimulation, which was previously thought a pre-requisite for efficient LNP transfection[41]. Our data suggest that the superior potency of LNP X is due to increased cellular association as well as an increased efficiency of mRNA translation after delivery to the cytosol, contrasting with earlier hypotheses that β-sitosterol enhanced the efficiency of endosomal escape[31]. Together, these findings identify LNP X as a promising in vitro transfection tool that has the potential to replace nucleofection or viral transduction in the development of therapeutics for a range of T cell-implicated diseases or the generation of T cell-based immunotherapies. Furthermore, our data justify exploring LNP X for delivering nucleic acid cargo to other hard-to-transfect cell types.

We here used LNP X to address the need for more potent, HIV-specific latency-reversing agents, in order to deplete the latent HIV reservoir and contribute to virologic control off ART[3]. Combined with an mRNA expressing the 72 amino acids of the first coding exon of HIV Tat, LNP X was able to overcome all blocks in HIV transcription initiation and RNA processing with striking potency, exceeding the viral reactivation achieved by stimulation with a general T cell mitogen. Tat-LNP X showed a greater ability to reverse HIV latency than previously reported Tat-LNPs[21,22,42]. The lack of disclosure of the LNP formulation and uncertainty of the full biological properties of the Tat$_{66}$ isoform used in the previously mentioned work means that direct comparisons of both Tat-LNPs were not possible. However, we hypothesise the superiority of our Tat-LNP X is primarily based on the high efficiency of mRNA delivery by LNP X, consistent with sub-optimal transfection efficiency of previously reported Tat-LNP[21]. Interestingly, the high level of latency reversal by Tat-LNP X appeared insufficient to drive virus-mediated cytotoxicity ex vivo, consistent with latently infected cells overexpressing pro-survival proteins such as B-Cell Lymphoma (BCL)-2[43,44] and BIRC-5[45]. These findings highlight the need to combine latency reversal with additional interventions to sensitise infected cells to death or enhance immune-mediated clearance[3,9,46]. Furthermore, single-cell or limiting-dilution assays will be required to assess the ability of Tat-LNP X to uniformly induce HIV transcription across the breadth of the HIV reservoir, including cells harbouring proviruses in a deep quiescent state[47,48]. We showed that LNP X could co-encapsulate and deliver CRISPRa machinery, including a large dCas9-VP64 mRNA. One limitation of our study was that the delivery efficiency of the three CRISPRa RNA components was not individually assessed. Direct analysis of the efficiency and kinetics of delivery of various RNA cargo, as well as their respective half-lives once delivered, are likely to be key in further optimisation of the CRISPR activation potency in primary cells[49]. Nonetheless, the potency of CRISPRa-LNP X is a promising indication that LNP X could be used to deliver other CRISPR variants to T cells, such as CRISPR-Cas9 to excise the integrated HIV provirus[50,51], or knock out CCR5 as the HIV entry receptor[52].

Our current results did not indicate any in vitro toxicity induced by treatment with LRA-LNP X, nor any signs of cellular activation. However, an in-depth analysis of changes to the cellular transcriptome or metabolome is warranted in further investigations into Tat or CRISPRa as an HIV-specific latency-reversing agent. To explore the potential of LNP X for in vivo therapeutics, studies assessing the immunogenicity, biodistribution and half-life in circulation will be required, as well as dose-finding and safety studies to determine the optimal therapeutic dose. Prior studies that administered LNP intravenously in people have shown that $0.1 - 0.6$ mg/kg of mRNA was well-tolerated in humans[2,53,54]. Future work exploring the potential to specifically target LNP X to T cells through ligand-mediated targeting could further contribute to the therapeutic potential of LNP X in vivo.

 

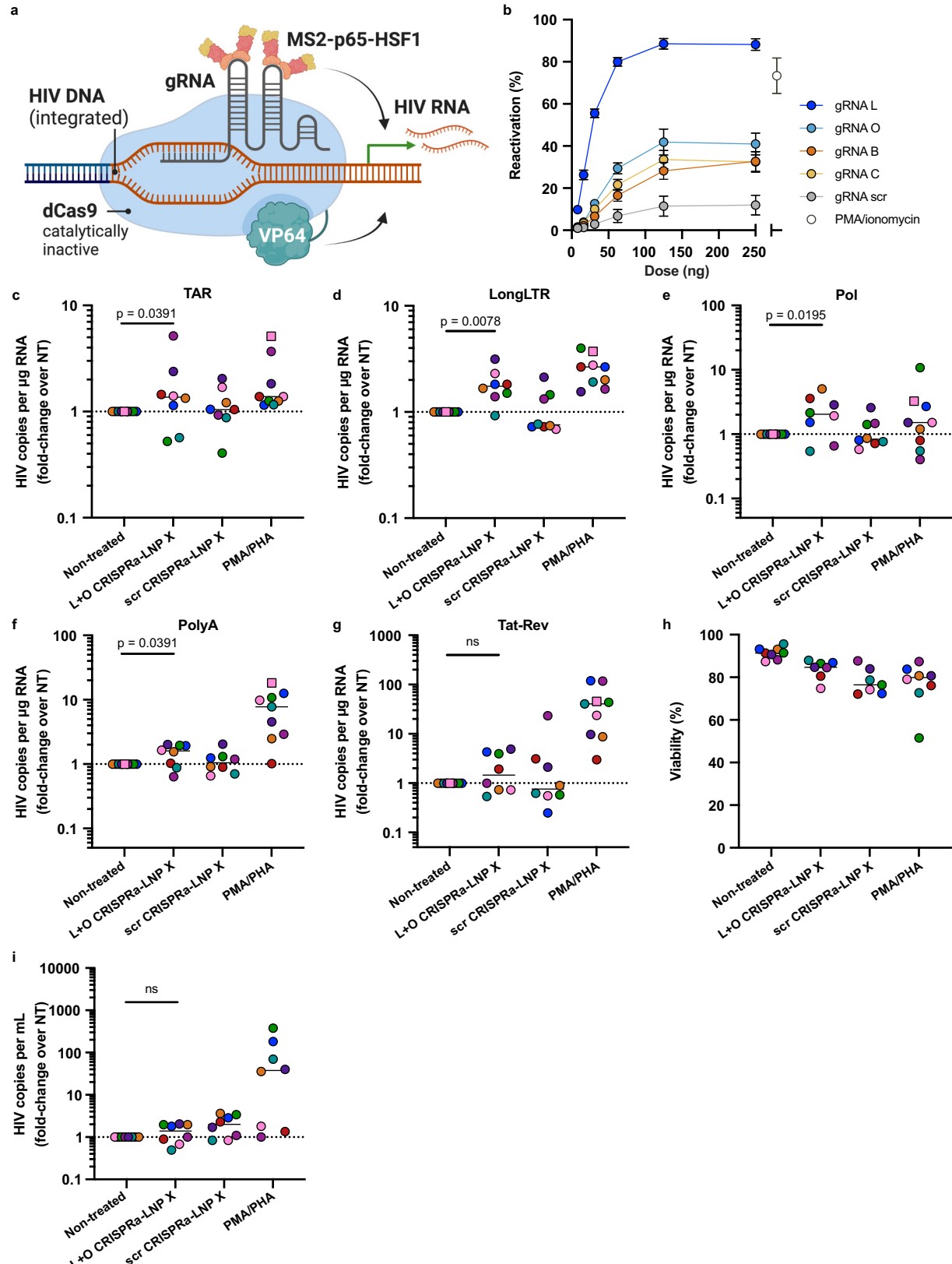

## Methods

### Ethics statement

Blood samples from people living with HIV were collected through leukapheresis at the Alfred Hospital (Melbourne, Australia) and the University of California San Francisco (UCSF) (USA) with informed consent and under institutional guidelines. People living with HIV included in these studies were aged 18 years or older and on suppressive antiretroviral treatment (ART) (viral load <50 copies/mL) for ≥3 years. The use of these samples was approved by the Human Research Ethics Committees at the Alfred Hospital, the University of Melbourne, and the Institutional Review Board at UCSF.

**Fig. 5 | CRISPRa-LNP X activates HIV transcription in CD4⁺ T cells from people living with HIV. a** Schematic overview of the CRISPRa system targeting the HIV LTR. Created in BioRender: *Cevaal, P. (2025)* https://BioRender.com/p8j24y6. **b** J-Lat 10.6 cells were treated for 24 h with indicated doses of CRISPRa-LNP X containing one of four HIV LTR-targeting gRNAs (L, O, B, C) or scrambled control gRNA. Reactivation of HIV LTR-mediated transcription was determined after 24 h by measuring GFP expression. Highest dose of CRISPRa-LNP X corresponds to 2.5 µg/mL. Treatment with PMA/ionomycin was included as a positive control. Mean ± SEM, *n* = 3 independent experiments. **c–i** CD4⁺ T cells from people living with HIV on suppressive ART were treated with 200 ng CRISPRa-LNP X containing gRNAs L and O (L + O CRISPRa-LNP X) or a scrambled gRNA control (scr CRISPRa-LNP X) per 10⁵ cells (4 µg/mL) or PMA/PHA as a positive control. After 48 h (squares) or 72 h (circles), expression of HIV transcripts TAR (**c**) LongLTR (**d**), Pol (**e**), PolyA (**f**) and Tat-Rev (**g**) representing transcription initiation, proximal elongation, distal elongation, completion and splicing, respectively, was determined using digital RT-PCR. Data were normalised to RNA input, then presented as fold-change induction compared to the corresponding non-treated (NT) control. Horizontal dashed line (**c–g**) represents no change relative to untreated cells. **h** Concurrently, cellular toxicity was determined using flow cytometry. Where datapoints are missing in (**h**), cell input was insufficient to perform an accurate measurement. **i** After 72 h, the number of copies of HIV RNA per mL of supernatant was quantified using RT-PCR and normalised to the non-treated control. **c–i** Short horizontal line represents the median of *n* = 7–8 donors. Significance in (**c–g**) and (**i**) was determined using a one-tailed Wilcoxon signed-rank test, ns non-significant.

## CD4⁺ T cell isolation and cell culture

Peripheral blood mononuclear cells (PBMCs) were obtained using Ficoll-Paque density gradient centrifugation of buffy coats (Australian Red Cross Lifeblood) or whole blood collected through leukapheresis. CD4⁺ T-cells were isolated from PBMCs using the EasyStep™ Human CD4⁺ T cell Isolation Kit (Stemcell Technologies), following manufacturer's instructions, and stored in liquid nitrogen until use. After thawing, CD4⁺ T cells were rested overnight prior to treatment and cultured in RPMI 1640 (Thermo Fisher Scientific) supplemented with 10% heat-inactivated foetal bovine serum (FBS; Cellsera Australia), 100 U/mL penicillin, 100 µg/mL streptomycin, 2 mM L-glutamine (all Thermo Fisher Scientific) (RF10) and 10 U/mL interleukin-2 (IL-2; Sigma-Aldrich). J-Lat A2 (RRID:CVCL_1G43) and J-Lat 10.6 cells (RRID:CVCL_8281)[55] were cultured in RF10 media. Cell lines and primary cells were maintained in a 37 °C humified incubator with 5% CO₂.

## RNA reagents for LNP synthesis

For studies comparing LNP X potency to benchmark LNP formulation (Fig. 1, Fig. S1), CleanCap® mCherry mRNA containing 5-methoxyuridine base modifications was obtained from TriLink Biotechnologies (L-7203). All other studies were performed with codon-optimised, capped, N1-methylpseudouridine base-modified mCherry mRNA purchased from Messenger Bio (Melbourne, Australia). Reporter mRNA encoding mScarlet was synthesised using in vitro transcription from codon-optimised PCR template using HiScribe T7 High Yield RNA Synthesis Kit (New England Biolabs), replacing all uridine with N1-methylpseudouridine (TriLink Biotechnologies). mRNA was co-transcriptional capped using CleanCap Reagent AG (TriLink Biotechnologies) and cleaned up by cellulose to remove dsRNA. AF488-oligo (5′-AF488AGA GTT CCC AAG ACC AGG CGG-3′) and Cy5-oligo (5′-Cy5AGA GTT CCC AAG ACC AGG CGG-3′) were purchased from Integrated DNA technologies. SNAP_switch-labelled oligo was synthesised by conjugating SNAP_switch-azide (ADKL Labs) to DBCO-oligo (5′-TCA GTT CAG GAC CCT CGG CTDBCO-3′) (TriLink Biotechnologies) as described previously[33]. Codon-optimised mRNAs encoding dCas9-VP64 (MW: 1528 kDa) and MS2-p65-HSF1 (MW: 562.6 kDa) were produced by Messenger Bio using previously published sequences[39]. mRNA expressing the 72 amino acids of the first coding exon of HIV Tat was designed following the NL4-3 reference sequence and produced by Messenger Bio (MW: 177.7 kDa). All custom-designed mRNAs purchased from Messenger Bio included N1-methylpseudouridine base modifications. Guide RNAs (gRNAs) were designed using previously described spacer sequences targeting the HIV LTR[23,40] (Table S1) or *CD25* (*IL2RA*) (5′-TTATGGGCGTAGCTGAAGAA-3′)[56] and a CRISPRa gRNA scaffold containing two MS2 hairpin aptamers, as described previously[39] (Table S1). gRNAs were obtained from Integrated DNA technologies using custom IDT Alt-R™ gRNA synthesis, containing two 2′O-methyl base and phosphorothioate bond modifications at the 5′ and 3′ end of the gRNA (MW: 52.0 kDa).

## Lipid nanoparticle formulation

Lipid nanoparticles were assembled using the ionisable lipid DLin-MC3-DMA (HY-112251, MedChemExpress) or SM-102 (33474, Cayman Chemical Company), DSPC (850365P, Avanti Polar Lipids), Cholesterol (C3045, Sigma-Aldrich) or β-sitosterol (700095P, Sigma-Aldrich), and DMG-PEG2000 (880151P, Avanti Polar Lipids). Lipids were reconstituted at 10 mM in ethanol and mixed at a molar ratio of 50:10:38.5:1.5 of DLin-MC3-DMA:DSPC:Cholesterol:DMG-PEG2000 or SM-102:DSPC:β-sitosterol:DMG-PEG2000 for patisiran LNPs and LNP X, respectively. Immediately prior to mixing, RNA was diluted to 150 ng/µL in 30 mM RNAse-free sodium acetate buffer, pH4.0. For CRISPRa-LNPs, RNA components were mixed at a mass ratio of 0.8:0.00625:1 of dCas9-VP64 mRNA: MS2-p65-HSF1 mRNA: gRNA, corresponding to a molar ratio of 0.027:0.00058:1. LNPs were synthesised using a NanoAssemblr Spark (Precision NanoSystems) whereby the aqueous (RNA) and organic (lipid) phases were mixed at a flow rate ratio of 1.8 (aqueous): 1 (organic), with an N/P ratio of 6:1. These formulation parameters are summarised in Table S3. The LNP size and polydispersity were analysed through dynamic light scattering using a Zetasizer Ultra (Malvern Panalytical). RNA encapsulation efficiency and total RNA concentration of the LNP formulations were determined using a Quant-it™ RiboGreen RNA Assay as per the manufacturer's low-range assay protocol (Invitrogen). Total RNA concentrations were used to dose LNPs throughout. For endosomal escape studies (Fig. 2), LNPs encapsulating a mixture of 95:5 (mass ratio) mScarlet mRNA and either SNAP_switch/AF488-labelled or Cy5/AF488-labelled oligos were synthesised using a NanoAssemblr Benchtop as described previously[33].

## Transfection of mRNA in CD4⁺ T cells in the presence and absence of other cell types

To activate CD4⁺ T cells prior to transfection, cells were stimulated with plate-bound anti-CD3 (clone OKT3; BioLegend) and 2 µg/mL of soluble anti-CD28 (clone CD28.2, Biolegend) for 72 h. In all studies using HIV-negative CD4⁺ T cells, 100,000 cells were treated with indicated doses of patisiran LNPs or LNP X in a total culture volume of 200 µL of RF10 with 10 U/ml IL-2. After 72 h, cells were stained using the LIVE/DEAD™ Fixable Violet Dead Cell Stain Kit (Thermo Fisher Scientific) and fixed prior to read-out. To identify CD4⁺ T cell subsets, the following antibodies were used: FITC anti-human CD3 (Clone UCHT1, BD Biosciences), BUV805 anti-human CD4 (Clone SK3, BD Biosciences), PerCP/Cy5.5 anti-human CD45RA (Clone HI100, BioLegend), APC/Cy7 anti-human CCR7 (Clone C043H7, BioLegend) and BV711 anti-human CD27 (Clone L128, BD Biosciences). For CD25 activation experiments, non-stimulated CD4⁺ T cells were incubated with CRISPRa-LNPs for either 72 h or 6 days and additionally stained with PE/Cyanine7 anti-human CD25 (Clone BC96; BioLegend). mCherry expression, CD25 expression and cell viability were assessed by flow cytometry using an LSRFortessa™ (BD Biosciences). To assess the tropism and specificity of LNP X, PBMCs from HIV-negative donors were plated at 300,000 cells in 100 µL of RF10 and treated with 200 ng mScarlet-LNP X for 24 h. PBMC cell subsets and the expression of mScarlet was determined by flow cytometry using the following antibodies: FITC anti-human CD14 (Clone M5E2, BD Biosciences), APC/Cy7 anti-human CD56 (Clone HCD56, BioLegend), BUV395 anti-human CD3 (Clone UCHT1, BD Biosciences), BUV805 anti-human CD8 (Clone SK1,

BD Biosciences), BV510 anti-human CD19 (Clone HIB19, BioLegend), BV650 anti-human HLA-DR (Clone G46-6, BD Biosciences), BV785 anti-human CD16 (Clone 3G8, BioLegend), PE/Cy7 anti-human CD4 (Clone OKT4, BioLegend).

## Quantifying LNP endosomal escape
LNP association, cytosolic delivery and expression of mRNA and endosomal escape efficiency of LNPs were determined in Jurkat T cells as described previously[33]. In brief, SNAP-actin expressing Jurkat T cells were treated with 100 ng mScarlet/SNAP$_{switch}$ oligo-LNPs per 100,000 cells for 4 h. Background-corrected geometric mean fluorescence was measured using an Aurora flow cytometer (Cytek) to assess LNP association (AF488), cytosolic delivery (SNAP$_{switch}$) and mRNA expression (mScarlet). To determine endosomal escape efficiency, the cytosolic delivery observed with the SNAP$_{switch}$-LNPs was compared to a reference LNP containing an unquenched Cy5-labelled oligo, representing an endosomal escape efficiency of 100%.

## J-Lat cell line reactivation
J-Lat A2[55] (ARP-9854) and J-Lat 10.6[55] (ARP-9849) cells were obtained through the AIDS Research and Reference Reagent Programme, Division of AIDS, NIAID, NIH. Cells were plated at 100,000 cells per well and treated with Tat-, mCherry- or CRISPRa-LNP X in a total culture volume of 100 μL RF10. For assessment of the optimal ratio of gRNA and CRISPRa mRNA components, LNPs containing either gRNA L, gRNA scr, dCas9-VP64 mRNA, or MS2-p65-HSF1 mRNA alone were used. For assessment of multiplexed gRNA delivery, CRISPRa-LNP X containing either gRNA L, gRNA O, or a combination of gRNA L and gRNA O (co-encapsulated at a 1:1 mass ratio) and CRISPRa mRNA components were used. J-Lat cells treated with phorbol 12-myristate 13-acetate (PMA, 16 nM, Sigma-Aldrich) and ionomycin (500 nM, Sigma-Aldrich) were used as positive controls. After 24 h, cells were stained with LIVE/DEAD™ Fixable Violet Dead Cell Stain Kit (Thermo Fisher Scientific). Reactivation of LTR-mediated transcription was measured by quantifying GFP expression using flow cytometry.

## Reactivation of HIV reporter virus
CD4$^+$ T cells from HIV-negative donors were spinoculated with pMorheus-V5[35] at a TCID$_{50}$ unit per cell of 0.005 in RF10 supplemented with 10 U/mL IL-2. pMorpheus-V5 is a single-round virus that harbours an mCherry reporter construct under the control of the HIV LTR promotor, allowing for the identification of productively infected cells undergoing LTR-mediated transcription. After 3 days, virus was aspirated and cells were seeded in fresh RF10 supplemented with 10 U/mL IL-2, then treated with 16 nM of PMA plus 500 nM ionomycin, 2 μg Tat-LNP X per million cells or 0.1% DMSO in a final culture volume of 1 mL per million cells. After a further 48 h, the levels of productive HIV infection were determined by quantifying mCherry expression within live cells.

## Ex vivo HIV reactivation
PBMCs from people living with HIV on suppressive ART were thawed, after which CD4$^+$ T cells were isolated using using the EasyStep™ Human CD4$^+$ T cell Isolation Kit (Stemcell Technologies), following manufacturer's instructions. Isolated CD4$^+$ T cells were rested overnight in RF10 containing 10 U/mL of IL-2 and 1 μM of Raltegravir. $2 \times 10^6$ cells were then treated with 4 μg of LNPs at a concentration of 4 μg/mL, or 10 nM of PMA and 10 μg/mL of phytohaemagglutinin (PHA) as a positive control. After 72 h, cells were separated from supernatant using centrifugation. Supernatant HIV RNA was quantified by the Victorian Infectious Diseases Reference Laboratory (VIDRL) using the Alinity m HIV-1 assay (Abbott) as previously described[57]. Cells were stained with LIVE/DEAD™ Fixable Violet Dead Cell Stain Kit and anti-CD25 (Clone BC96, BioLegend), Brilliant Ultra Violet™ 395 anti-CD69 (Clone FN50, Invitrogen) and FITC anti-HLA-DR (Clone L243, Invitrogen). Cell viability and the expression of cellular activation markers were assessed using flow cytometry. To quantify the induction of HIV transcripts, RNA was extracted using polyacryl carrier enriched TRI Reagent (Molecular Research Center) according to the manufacturer's protocol. TAR levels were quantified by 3-step polyadenylation-RT-dPCR, whilst Long-LTR, Pol, PolyA and Tat-Rev transcripts were quantified by 2-step RT-dPCR using the QIAcuity Four 5-plex digital PCR system (Qiagen), as previously described[36]. HIV RNA copies were normalised to RNA input. To quantify changes to the size of the proviral reservoir, cells were treated with Tat-LNP X for 120 h. DNA was extracted using AllPrep DNA/RNA Mini Kit (Qiagen) as per the manufacturer's protocol. Intact, 5' defective and 3' defective HIV DNA was quantified using Qiagen's QIAcuity Four 5-plex digital PCR targeting parts of the *psi* and *env* region, as previously described[58]. The RPP30 host gene was used to measure the analysed number of cells and the DNA shearing index (DSI) as an indication of DNA fragmentation. The number of HIV copies was corrected to DSI and normalised to cell input.

## Statistics
Statistical analyses were performed in Graphpad Prism 10. A Student's t test or ratio paired t test was used to compare two groups for primary cell transfection experiments involving uninfected cells or cells infected in vitro with pMorpheus-V5. For ex vivo experiments using CD4$^+$ T cells from people living with HIV, a one-sided (for induction of HIV RNA) or two-sided (for changes to intact proviral DNA levels) Wilcoxon signed-rank test was used to compare treatment conditions with untreated or positive controls.

## Reporting summary
Further information on research design is available in the Nature Portfolio Reporting Summary linked to this article.

# Data availability
Source data are provided for all experimental results presented in the main manuscript and supplementary information. Source data are provided with this paper.

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

## Acknowledgements

We would like to thank the people living with HIV for generously donating blood to this study. We would like to thank the Infectious Diseases Unit at the Alfred Hospital and the Lewin Clinical Research Group at the Peter Doherty Institute for providing the leukapheresis samples. Specific thanks to Ajantha Rhodes, Judy Chang, Ashanti Dantanarayana, Sureka Tennakoon, Socheata Chea and Lauren Wallace for processing, storing and managing the leukapheresis samples. We acknowledge Professor Viviana Simon and Assistant Professor Lubbertus Mulder from the Icahn School of Medicine at Mount Sinai for supplying plasmids required for the production of the pMorpheus-V5 reporter virus. This research was supported by the Australian National Health and Medical Research Council (NHMRC; Programme Grant No. GNT1149990, Practitioner Fellowship No. 1135851, Investigator Grant No. 2026490 (SRL), Investigator Grant No. GNT2016732 (FC) and The Medical Research Future Fund Grant No. MRF2016144), the National Institute of Allergy and Infectious Diseases Delaney AIDS Research Enterprise to find a cure (DARE) (1UM1AI164560-01), the Australian Centre for HIV and Hepatitis Virology Research (ACH4), the Harold & Cora Brennen Benevolent Trust (Equipment grant No. BREN20122), The Foundation for AIDS Research (amfAR; TARGET Grant No. 110406-73-RGRL_Lewin), The Gandel Foundation and The mRNA Victoria Research Acceleration Fund (mVRAF, GA-F3673757-3598). S.T. is supported by the Doherty Institute for Infection and Immunity Locarnini Fellowship in Virology and University of Melbourne Department of Infectious Diseases Support Package. The authors acknowledge the Melbourne Cytometry Platform (Peter Doherty Institute node) for the provision of flow cytometry services. Schematic figures were generated using Biorender.com.

## Author contributions

P.M.C., F.C., J.S., S.R.L. and M.R. conceived the project. P.M.C., B.M.F., M.A.M., A.T., K.T., R.A.S., H.L., A.P.R.J., S.R.L. and M.R. designed the experiments. P.M.C., B.M.F., M.A.M., A.T., K.T., R.A.S., Y.K., J.O. and H.L. executed experiments. S.K., A.A., D.L.F., M.H., D.F.J.P., J.M.L.C., M.F., T.P., W.Z., S.T., C.P. and F.C. provided intellectual input in the experimental design and the design and synthesis of mRNA and lipid nanoparticles. S.K. conceived the LNP X formulation. P.M.C., B.M.F., M.A.M., A.T., K.T., R.A.S. and H.L. performed the experimental analysis. P.M.C., B.M.F., M.A.M., K.T., R.A.S., H.L., A.P.R.J., S.R.L. and M.R. interpreted the data. J.H.M., S.G.D., R.H. and S.R.L. provided access to leukapheresis samples of people living with HIV on ART. P.M.C., B.M.F., M.A.M., S.R.L. and M.R. wrote and edited the manuscript. P.M.C. prepared the figures. All authors read and approved the final manuscript.

## Competing interests

S.R.L. has received honoraria unrelated to the content of this manuscript for participation in advisory boards for Gilead, Viiv Healthcare, Merck, Abbvie, Esfam, Immunocore and First Health. She has received funding from Gilead and Merck for investigator-initiated research projects unrelated to the content of this manuscript. DLF and FC are scientific founders of and hold equity in Messenger Bio. Messenger Bio was paid to synthesise mRNA for 60% of the work presented in the manuscript. P.M.C. was supported by a donation from the J and M Wright Foundation. The donation supported salary to complete some of this work. B.M.F., M.A.M., R.A.S. were supported by an Australian Government Research Training Programme Scholarship. M.A.M. was further supported by an NHMRC Postgraduate Scholarship, Rowden White Scholarship and Elizabeth Mary Sweet Scholarship. K.T. was supported by a Melbourne Research Scholarship. P.M.C., S.K., B.M.F., M.A.M., M.F., M.R., J.S. and S.R.L. are named investigators on a patent related to this work (PCT/AU2024/050506). No other authors declare any competing interests.
