## [Peer Review file · Nature Communications]

Efficient mRNA delivery to resting T cells to reverse HIV latency

Corresponding Author: Professor Sharon Lewin

Version 0:

Reviewer comments:

Reviewer #1

(Remarks to the Author)

Cevaal and colleagues have studied the delivery of LipidX nanoparticles carrying HIV Tat mRNAs or synthetic gene activators targeted by CRISPr to the HIV promoter, resulting in expression of HIV. Overall, the manuscript is clear and the findings are of general interest. In addition to evidence of the induction of proviral RNA expression, in Figure 3i they have provided some preliminary evidence of latency reversal that progresses to the ultimate level, resulting in viral RNA release from infected cells.

The fold induction of proviral expression is generally robust, but as such levels of expression are often low in absolute measures, fold changes can often over-represent the findings. It was a bit difficult to be sure, but it seems that a supplemental Excel file labelled Fig 5S a-e is the raw RNA data corresponding to the absolute changes shown in Fig. 3a-e. If so, this demonstrates that the fold increases seen are in most cases not artifactually inflated by very low baseline levels, as the basal levels of expression are indeed quite robust, and the fold changes quite consistent between samples.

In our experience with HIV+ donor cells this is quite unusual, as we find basal expression in freshly isolated resting CD4 cells to be often very low, and also may vary significantly from donor to donor. We would suggest that the authors revisit their methods, the details of which were difficult to ascertain. We generally rest cells for a few hours in low concentrations of IL2 after they have been exposed to the gravimetric stresses of purification, and then re-confirm surface marker status.

All previous studies delivering some form of HIV Tat have also shown an efficacy in latency reversal that is variable between donors. Strikingly and uniquely in this cohort the effect is strikingly uniform. The cause of variable efficacy of Tat prior studies has been unclear, and it has not been determined if either a) Tat is not a critical limiting factor driving proviral quiescence in some latently infected cells, b) the delivery of Tat by previously published reagents was inconsistent, or c) both factors contribute. This is of critical importance if we are to understand if Tat is really the "perfect" reagent to reverse latency, or simply a new (but important) tool. Here the authors seem to claim that Lipid X provides uniquely efficient and uniform delivery, and that Tat is a potent and uniformly active latency reversal agent. This might be correct, but further studies are needed to be certain of this. I hope that further studies in primary cells and in vivo can validate this optimistic finding. It might be wise for the authors to be circumspect in their conclusions for now.

Reviewer #2

(Remarks to the Author)

The authors have addressed our concerns, and I now recommend publication.

Reviewer #3

(Remarks to the Author)

Here, the authors describe a novel lipid nanoparticle (LNP) that can transfect mRNA in vitro into human CD4 T cells, even when they are not activated (referred to as 'resting' in the article). They use these LNPs to deliver mRNA encoding HIV Tat

or a CRISPR activation machinery that can reactivate endogenous HIV viruses. It is an interesting finding that a new LNP (having an alternative ionisable lipid and sterol compared to the commercialised Patisiran) can transfect CD4 T-cells in vitro. However, the required dose is high (2.5 to 4 micrograms per ml) and could not be reached in vivo (injecting 20 mg mRNA for 5 liters of blood?). The formulation is not specific for T-cells: all other blood cell types are also transfected (is the bar for B-cells correct in Fig S2 f?). Most importantly, "the high level of latency reversal by Tat-LNP X appeared insufficient to drive virus-mediated cytotoxicity ex vivo" thus it is not clear how the method would help against HIV infection. There are no in vivo data. Thus, the method presented (an LNP that can transfect mRNA in 'resting' CD4 T cells in vitro) is interesting (for example using LNPs to deliver mRNA encoding chimeric antigen receptors). However, the facts that the strategy does not induce virus-mediated cytotoxicity in vitro and they have not tested in vivo whether this technology can work might make this study still a bit preliminary for a very high ranking journal. It cannot be deduced from the results presented that this method could work in vivo and be of interest for medical use (compared, for example, to targeted LNPs delivering mRNA to specific cell types as described, for example, by Rurik et al 'CAR T cells produced in vivo to treat cardiac injury' Science. 2022 Jan 7;375(6576)).

Reviewer #4

(Remarks to the Author)

The authors have not conducted any additional experiments that I originally suggested, but the manuscript is otherwise improved and, in my opinion, is a good fit for Nat Com. I have no further comments.

REVIEWER COMMENTS

Reviewer #1 (Remarks to the Author):

Cevaal and colleagues have studied the delivery of LipidX nanoparticles carrying HIV Tat mRNAs or synthetic gene activators targeted by CRISPr to the HIV promoter, resulting in expression of HIV. Overall, the manuscript is clear and the findings are of general interest. In addition to evidence of the induction of proviral RNA expression, in Figure 3i they have provided some preliminary evidence of latency reversal that progresses to the ultimate level, resulting in viral RNA release from infected cells.

The fold induction of proviral expression is generally robust, but as such levels of expression are often low in absolute measures, fold changes can often over-represent the findings. It was a bit difficult to be sure, but it seems that a supplemental Excel file labelled Fig 5S a-e is the raw RNA data corresponding to the absolute changes shown in Fig. 3a-e. If so, this demonstrates that the fold increases seen are in most cases not artifactually inflated by very low baseline levels, as the basal levels of expression are indeed quite robust, and the fold changes quite consistent between samples.

This is correct. The raw RNA measures of proviral expression are graphed in Figure S6a-e of the revised manuscript. We have attached an updated source data file that shows the numerical values graphed in Fig 3 (fold-change) and Fig S6 (raw values).

In our experience with HIV+ donor cells this is quite unusual, as we find basal expression in freshly isolated resting CD4 cells to be often very low, and also may vary significantly from donor to donor. We would suggest that the authors revisit their methods, the details of which were difficult to ascertain. We generally rest cells for a few hours in low concentrations of IL2 after they have been exposed to the gravimetric stresses of purification, and then re-confirm surface marker status.

CD4⁺ T cells were isolated using a negative selection kit to avoid non-specific activation and rested overnight in 10 units/mL of IL-2 prior to treatment. We have clarified this methodology in lines 392-395 of the Methods section:

“PBMCs from people living with HIV on suppressive ART were thawed, after which CD4⁺ T cells were isolated using using the EasyStep™ Human CD4⁺ T cell Isolation Kit (Stemcell Technologies), following manufacturer’s instructions. Isolated CD4⁺ T cells were rested overnight in RF10 containing 10 U/mL of IL-2 and 1 μM of Raltegravir. 2x10⁶ cells were then treated with 4 μg of LNPs at a concentration of 4 μg/mL, or 10 nM of PMA and 10 μg/mL of phytohaemagglutinin (PHA) as a positive control.”, line 392-395

Using this protocol and even after treatment with LNP X, ex vivo isolated CD4⁺ T cells retained their resting phenotype (see Figures S6f-h and S10f-h). Furthermore, in isolated CD4⁺ T cells from HIV-negative donors, we detected baseline expression of CD25 (an early T cell activation marker) in <10% of cells (Figure 4c), again confirming that the freeze/thaw and negative isolation procedure did not result in activation of the cells.

With regards to the level of cell associated HIV RNA that we measure using RNA transcription profiling, the levels of TAR ($10^4 - 10^5$ copies/ μg RNA); LongLTR (10^3 - 10^4 copies/ μg RNA); Pol (10^3 - 10^4 copies/ μg RNA); PolyA (10^3 - 10^4 copies/ μg RNA); and Tat-Rev (1 - 10^2 copies/ μg RNA) is consistent with what we have previously seen in uncultured CD4+ T cells (Tumpach *et al.*, Viruses (2023), PMID: 37515292) and what has been seen by the Yukl lab who pioneered the HIV RNA transcription profiling assay (Yukl *et al.*, Sci Transl Med (2018), PMID: 29491188, Figure 1). Please note that the denominator for the measurements of transcripts is per μg RNA, not copy number of another house keeping gene such as 18S RNA, as is sometimes used by others, including our own group in the past (Elliott et al Plos Path 2014; Elliott et al Lancet HIV 2018).

All previous studies delivering some form of HIV Tat have also shown an efficacy in latency reversal that is variable between donors. Strikingly and uniquely in this cohort the effect is strikingly uniform. The cause of variable efficacy of Tat prior studies has been unclear, and it has not been determined if either a) Tat is not a critical limiting factor driving proviral quiescence in some latently infected cells, b) the delivery of Tat by previously published reagents was inconsistent, or c) both factors contribute. This is of critical importance if we are to understand if Tat is really the “perfect” reagent to reverse latency, or simply a new (but important) tool. Here the authors seem to claim that Lipid X provides uniquely efficient and uniform delivery, and that Tat is a potent and uniformly active latency reversal agent. This might be correct, but further studies are needed to be certain of this. I hope that further studies in primary cells and in vivo can validate this optimistic finding. It might be wise for the authors to be circumspect in their conclusions for now.

We thank the reviewer for raising this interesting and highly relevant discussion. As highlighted by the reviewer, we observe relative uniformity in responses observed between donors. We did not identify donors resistant to Tat-LNP X-mediated latency reversal. We agree the cause of the donor variation observed in Van Gulck *et al.* and Raines *et al.* cannot be concluded from those publications, given 1) the absence of experiments confirming whether the non-responsiveness is a stochastic or a specific donor-dependent effect; and 2) the absence of data on the transfection efficiency of their LNP formulation.

We agree that further studies are needed to quantify the response of individual cells or individual proviruses to Tat-mediated latency reversal. We have added further discussion on this limitation into the revised manuscript, including a reference to the recently accepted review article on Tat from our group by Fisher *et al.* (Frontiers in Immunology, *in press*):

“Furthermore, single-cell or limiting-dilution assays will be required to assess the ability of Tat-LNP X to uniformly induce HIV transcription across the breadth of the HIV reservoir, including cells harbouring proviruses in a deep quiescent state^{47,48}.”, line 267-269

Reviewer #2 (Remarks to the Author):

The authors have addressed our concerns, and I now recommend publication.

Reviewer #3 (Remarks to the Author):

Here, the authors describe a novel lipid nanoparticle (LNP) that can transfect mRNA *in vitro* into human CD4 T cells, even when they are not activated (referred to as 'resting' in the article). They use these LNPs to deliver mRNA encoding HIV Tat or a CRISPR activation machinery that can reactivate endogenous HIV viruses. It is an interesting finding that a new LNP (having an alternative ionisable lipid and sterol compared to the commercialised Patisiran) can transfect CD4 T-cells *in vitro*. However, the required dose is high (2.5 to 4 micrograms per ml) and could not be reached *in vivo* (injecting 20 mg mRNA for 5 liters of blood?).

Based on a number of recent clinical studies, an intravenous dose of mRNA-LNP of 0.1 – 0.6 mg/kg appears to be acceptable and well-tolerated. For example, Gillmore *et al.*, NEJM (2021) (PMID: 34215024) observed only mild adverse events in patients dosed 0.3 mg/kg i.v.; August *et al.*, Nature Med (2021) (PMID: 34887572) reported no serious adverse events in single intravenous doses up to 0.6 mg/kg or two weekly doses of 0.3 mg/kg; Koeberl *et al.*, Nature (2024) (PMID: 38570682) reported no dose-limiting toxicities for doses up to 0.9 mg/kg i.v. every 2 weeks for up to 10 doses.

For an adult weighing 80 kgs, 0.1 – 0.6 mg/kg would equal a total dose of 8 – 48 mg of mRNA-LNP, suggesting that an intravenous injection of 20 mg mRNA would be feasible. We furthermore believe that the required dose can be reduced if LNP X is functionalized with T-cell targeting moieties. The safety and tolerability of LNP X, or any derivative, will need to be carefully studied in animal models and Phase I trials, but those studies are beyond the scope of this manuscript.

We agree that the *in vivo* therapeutic potential of LNP X is important to discuss in more detail, and have updated the Discussion to add further clarification:

“To explore the potential of LNP X for *in vivo* therapeutics, studies assessing the immunogenicity, biodistribution and half-life in circulation will be required, as well as dose-finding and safety studies to determine the optimal therapeutic dose. Prior studies that administered LNP intravenously in people has shown that 0.1 – 0.6 mg/kg of mRNA was well-tolerated^{2,53,54}. Future work exploring the potential to specifically target LNP X to T cells through ligand-mediated targeting could further contribute to the therapeutic potential of LNP X *in vivo*.”, line 280-285

The formulation is not specific for T-cells: all other blood cell types are also transfected (is the bar for B-cells correct in Fig S2 f?).

The reviewer is correct in that LNP X does not have an inherent specificity for T cells, though our findings demonstrate a high tropism for non-stimulated T cells. We describe this in lines 119-122:

“Even in the context of peripheral mononuclear cells (PBMCs), LNP X was able to deliver mRNA to CD4⁺ T cells (Figure S2e,f). Expression of mRNA was also detected in most other PBMC subsets, specifically monocytes, demonstrating that LNP X is T cell tropic, but not T cell specific.”, lines 119-122

The bar for B cells in Fig S2f is correct; we consistently observed minimal transfection of B cells when treating PBMCs with LNP X. The findings in Fig S2f represent the mean \pm SEM of n=6 donors, who were treated across 3 separate experiments using 3 separate LNP X preparations. The low transfection efficiency of B cells within PBMCs is consistent with recent findings published by Chen *et al.*, ACS Bio Med Chem Au (2024) (PMID: 39990935) across a range of LNP formulations.

We mentioned in the discussion that, in order to achieve a higher degree of specificity towards T cells, targeting using ligands conjugated to the LNP surface will likely be required:

“Further studies are also required to explore the potential of LNP X for *in vivo* therapeutics by assessing the immunogenicity, biodistribution and half-life in circulation, as well as the potential to specifically target LNP X to T cells through ligand-mediated targeting.”, line 277-279

Most importantly, "the high level of latency reversal by Tat-LNP X appeared insufficient to drive virus-mediated cytotoxicity ex vivo" thus it is not clear how the method would help against HIV infection. There are no in vivo data. Thus, the method presented (an LNP that can transfect mRNA in 'resting' CD4 T cells in vitro) is interesting (for example using LNPs to deliver mRNA encoding chimeric antigen receptors). However, the facts that the strategy does not induce virus-mediated cytotoxicity in vitro and they have not tested in vivo whether this technology can work might make this study still a bit preliminary for a very high ranking journal. It cannot be deduced from the results presented that this method could work in vivo and be of interest for medical use (compared, for example, to targeted LNPs delivering mRNA to specific cell types as described, for example, by Rurik et al 'CAR T cells produced in vivo to treat cardiac injury' Science. 2022 Jan 7;375(6576)).

The reviewer is correct that, in our hands, treatment with Tat-LNP X did not result in a decline in the HIV reservoir *ex vivo*. We believe that this is an extremely important finding for the field. As discussed, in the manuscript our findings demonstrate that latency reversal alone does not deplete the reservoir, even with a very potent stimulus such as Tat. However, we believe that the major goal of latency reversal is to induce expression of viral proteins to either sensitize infected cells to death or enhance immune-mediated clearance (Deeks *et al.*, Nat Med (2021); Kim *et al.*, Cell Host Microbe (2018); Dashti *et al.*, Nat Med (2023)). Therefore, an increase in antigen presentation remains an important component of a combination cure strategy.

Experiments to optimise the delivery of LNP-X in an animal model are currently ongoing and we believe are beyond the scope of this publication. We have

demonstrated a very novel finding of highly efficient transfection of resting CD4+ T-cells that we believe is of high importance in the HIV field and beyond.

Reviewer #4 (Remarks to the Author):

The authors have not conducted any additional experiments that I originally suggested, but the manuscript is otherwise improved and, in my opinion, is a good fit for Nat Com. I have no further comments.